# Antibody Fab-Fc properties outperform titer in predictive models of SIV vaccine-induced protection

Srivamshi Pittala[1], Kenneth Bagley[2], Jennifer A Schwartz[2], Eric P Brown[3], Joshua A Weiner[3], Ilia J Prado[2], Wenlei Zhang[2], Rong Xu[2], Ayuko Ota-Setlik[2], Ranajit Pal[4], Xiaoying Shen[5], Charles Beck[6], Guido Ferrari[6], George K Lewis[7] (iD), Celia C LaBranche[6], David C Montefiori[8], Georgia D Tomaras[5], Galit Alter[9,10], Mario Roederer[11], Timothy R Fouts[2,†], Margaret E Ackerman[3] (iD) & Chris Bailey-Kellogg[1,*] (iD)

## Abstract

Characterizing the antigen-binding and innate immune-recruiting properties of the humoral response offers the chance to obtain deeper insights into mechanisms of protection than revealed by measuring only overall antibody titer. Here, a high-throughput, multiplexed Fab-Fc Array was employed to profile rhesus macaques vaccinated with a gp120-CD4 fusion protein in combination with different genetically encoded adjuvants, and subsequently subjected to multiple heterologous simian immunodeficiency virus (SIV) challenges. Systems analyses modeling protection and adjuvant differences using Fab-Fc Array measurements revealed a set of correlates yielding strong and robust predictive performance, while models based on measurements of response magnitude alone exhibited significantly inferior performance. At the same time, rendering Fab-Fc measurements mathematically independent of titer had relatively little impact on predictive performance. Similar analyses for a distinct SIV vaccine study also showed that Fab-Fc measurements performed significantly better than titer. These results suggest that predictive modeling with measurements of antibody properties can provide detailed correlates with robust predictive power, suggest directions for vaccine improvement, and potentially enable discovery of mechanistic associations.

**Keywords** antibody effector function; biomarker identification; HIV; protection modeling; systems serology

**Subject Categories** Immunology; Methods & Resources; Microbiology, Virology & Host Pathogen Interaction

**Mol Syst Biol. (2019) 15: e8747**

## Introduction

While vaccine efficacy is often distilled into a single-measure, antibody titer (Plotkin, 2010; Ohmit *et al*, 2011; Katzelnick *et al*, 2016), the magnitude of the humoral response may not suffice to evaluate or explain outcomes; rather, qualitative aspects may be critically important (Abraham *et al*, 2002; Feng *et al*, 2009; Corey *et al*, 2015). For example, a recent study of two different RTS,S-based malaria vaccines found both to be equally protective despite having very different titer levels, leading to conjecture that antibody quantity was compensated by quality (Kazmin *et al*, 2017). Similarly, a study of three different Ebola vaccines revealed that while they induced similar levels of anti-Ebola antibodies, only one yielded a substantially better response providing total protection (Blaney *et al*, 2013). Additionally, a recent SIV vaccine study identified a number of specific antibody qualities that were associated with protection whereas IgG response magnitudes were not (Ackerman *et al*, 2018). Antibody quality has been frequently characterized in terms of neutralization potency and breadth (Mascola & Haynes, 2013; Hraber *et al*, 2014), as well as specificity to particular epitopes (Wrammert *et al*, 2011; Klein *et al*, 2013; Zolla-Pazner *et al*, 2014; Frei *et al*, 2015; Ha *et al*, 2017). Beyond these, antibody qualities such as opsonophagocytic activity, complement-dependent cytotoxicity, and NK cell-mediated cytolysis have often been assessed. These activities are strongly influenced by properties of the Fc domain,

1   Department of Computer Science, Dartmouth, Hanover, NH, USA
2   Profectus BioSciences, Inc., Baltimore, MD, USA
3   Thayer School of Engineering, Dartmouth, Hanover, NH, USA
4   Advanced Bioscience Laboratories, Inc., Rockville, MD, USA
5   Duke Human Vaccine Institute, Durham, NC, USA
6   Department of Surgery, Duke University Medical Center, Durham, NC, USA
7   Institute for Human Virology, University of Maryland School of Medicine, Baltimore, MD, USA
8   Duke University, Durham, NC, USA
9   Harvard Medical School, Boston, MA, USA
10  Ragon Institute of Massachusetts General Hospital, Massachusetts Institute of Technology and Harvard University, Boston, MA, USA
11  Vaccine Research Center, NIAID, NIH, Bethesda, MD, USA
    *Corresponding author. Tel: +1 603 646 3385; E-mail: cbk@cs.dartmouth.edu
    †Present address: Advanced Bioscience Laboratories, Inc., Rockville, MD, USA

including isotype, subclass, and glycan, suggesting the importance of these factors in driving suitable effector function (Barouch *et al*, 2015; Gordon *et al*, 2016; Huang *et al*, 2016). Collectively, there is a rich history of such qualitative antibody characteristics serving as important correlates for control of natural infection and vaccine efficacy (Johnson *et al*, 1999; Osier *et al*, 2008; Gómez Román *et al*, 2014; Weber & Oxenius, 2014; Ackerman *et al*, 2016; Zhong *et al*, 2016; He *et al*, 2017; Jegaskanda *et al*, 2017; Li *et al*, 2017; Rouers *et al*, 2017). In general, in addition to better distinguishing observed differences in immunity, expanded biophysical analyses of antibody properties can provide a more refined understanding of characteristics important for potent responses and potentially how to improve them.

Systems approaches provide the opportunity to broadly characterize immune responses and efficiently identify properties associated with protection and differences between vaccine compositions or immunization regimens (Querec *et al*, 2009; Kuri-Cervantes *et al*, 2016). While gene expression profiles are generally at the heart of systems biology, immunology-specific approaches have been developed to measure and leverage biological profiles (e.g., cytokines, multi-dimensional cellular markers, metabolites) (Furman & Davis, 2015; Lin *et al*, 2015; Davis *et al*, 2017). In order to enable "systems serology" studies (Chung *et al*, 2015; Ackerman *et al*, 2017), we have recently developed a platform, the "Fc Array", to comprehensively dissect antibody profiles of serum samples (Brown *et al*, 2012, 2017). By characterizing antibodies in terms of simultaneous Fab and Fc properties (i.e., both antigen specificity as well as subclass and ability to bind complement and different FcγR receptors), the Fc Array has provided a refined characterization of immune responses in a variety of vaccination and natural infection studies (Lai *et al*, 2014; Choi *et al*, 2015; Ackerman *et al*, 2016, 2018; Vaccari *et al*, 2016; Bradley *et al*, 2017).

Here, we reconsider a recent non-human primate vaccine study evaluating the efficacy of an intramuscular DNA immunization followed by a protein boost (Fouts *et al*, 2015) (Fig 1A). The DNA prime consisted of a plasmid DNA expressing rhesus full-length single chain (rhFLSC$_{smE660}$), an immunogen comprised of SIVsmE660 gp120 envelope glycoprotein fused to the rhesus macaque CD4 D1D2 domain, and a plasmid DNA encoding SIVsmE543 Gag and Pol. In addition to these HIV antigen-expression plasmids, groups of animals were treated with an additional plasmid expressing none, either, or both of the two genetic adjuvants: IL-12 and the catalytic A1 subunit of *E. coli* heat labile toxin (LTA1). Following three DNA immunizations at weeks 0, 4, and 8, responses were boosted by intramuscular vaccination with recombinant rhFLSC$_{CCG7V}$ protein (i.e., rhesus full-length single chain for a different SIV strain) in an aluminum adjuvant at week 44. Repeated low-dose rectal challenge with a heterologous SIVmac251 strain initiated at 2 weeks post-boost demonstrated that animals immunized with IL-12-adjuvanted DNA had significantly lower infection rates compared to animals in all the other groups (Fouts *et al*, 2015). Previous analysis identified a balance between strong antibody effector function, particularly antibody-dependent cellular cytotoxicity (ADCC), and relatively low cellular responses, as assessed by IFNγ expression of stimulated T cells, to be associated with protection (Fouts *et al*, 2015).

Seeking to further dissect the humoral component of the immune response in this study, Fc Array data were collected for serum samples on the day of first challenge and used to systematically investigate the ability of computational models to distinguish adjuvant groups, and predict and compare protection using (i) full antibody profiles to which titer may have contributed, (ii) titer-based measurements alone, and (iii) antibody profiles from which the contribution of titer has been mathematically removed (Fig 1B and C). Beyond defining new correlates of protection, the resulting analysis showed that the characterization of antibody qualities can be necessary to accurately model protection. Models based on characterizations of antibody Fab and Fc properties, either with or without an implicit titer component, significantly and substantially outperformed models using response magnitudes alone for differentiating group-based responses and the extent of protection. Identifying such properties provides the opportunity to hypothesize mechanisms of protection that may be tested in further studies.

## Results

### Antibody profiles

Thirty-two serum samples (*n* = 8/group, differing by the adjuvants used in the DNA priming immunizations, i.e., no DNA adjuvant (Empty); LTA1; IL-12; and the combination IL-12 + LTA1) were profiled on the Fc Array. The study also included 8 unvaccinated control animals, which were not considered here for modeling, but were instead used to set a baseline for signal in the experimental data. Since the IL-12 group was previously found to be significantly more protected compared to the other adjuvant groups, analyses here compared the IL-12 group to the other three adjuvant groups combined.

Each animal's antibody response was characterized in terms of two types of measurements, one which we call "titer features", measuring the overall magnitude of a particular antigen-specific response, and one which we call "Fc Array features", further characterizing the antigen-specific response in terms of antibody qualities, namely Fc properties. Both types of measurements are numerical in nature, and "qualitative" indicates that a measurement characterizes antibody qualities beyond magnitude. In fact, all data were collected simultaneously using the Fc Array, which multiplexes the characterization of Fab properties and Fc properties. Here, the Fc Array was used with reagents evaluating twelve antigen specificities (one for Pol, two for Gag, and nine for Env: gp120-140s and FLSC sequence variants comprising a variety of SIV strains) (Appendix Fig S1A), along with eight different interactions of the Fc domains of these antigen-specific antibodies with Fc receptors (Appendix Fig S1B, "Response Quality") and two anti-IgG detection conditions (Appendix Fig S1B, "Response Magnitude"). Each simultaneous Fab:Fc measurement is termed a "feature". The anti-IgG detection reagent measurements have been shown to correlate well with standard ELISA-based measurements of titer (Brown *et al*, 2012) and thus were used here as the quantitative titer features characterizing response magnitude, while the rest comprised the qualitative Fc Array features characterizing response properties. For modeling purposes, features were discarded if they displayed no statistically significant difference compared to

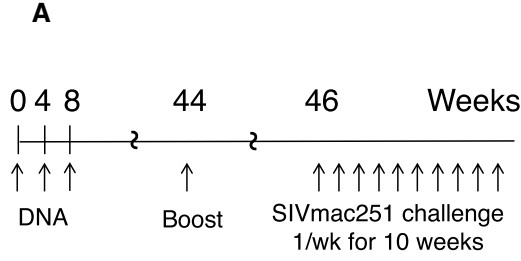

| Group | | DNA Adjuvant Plasmid | DNA Vaccine Plasmid | Boost |
|---|---|---|---|---|
| Others | | Empty | rhFLSC(smE660) + gag/pol(smE543) | rhFLSC$_{CCG7V}$ |
| | | LTA1 | | |
| | | LTA1 + rhIL-12 | | |
| IL-12 | | rhIL-12 | | |

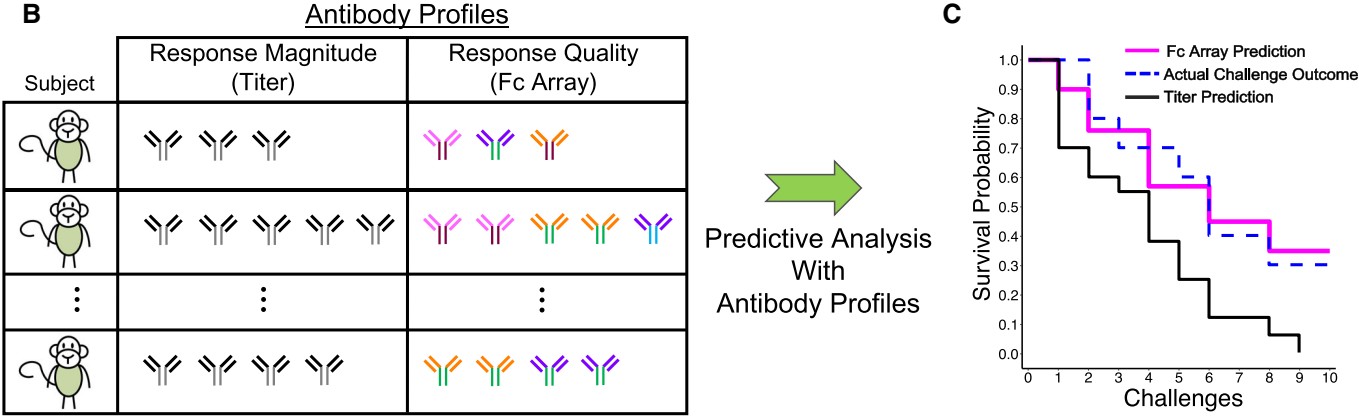

**Figure 1. Schematic overview for comparison of the predictive power of antibody response magnitude versus response quality data.**

A  Overview of the vaccine regimen (adapted from Fouts *et al*, 2015).

B  Antibody profiles of subjects are assayed, post-vaccination but pre-challenge, capturing either response magnitude alone (quantity, titer) or Fab and Fc properties (quality, Fc Array). These profiles are input into a predictive analysis framework that trains models to predict a subject's risk of infection based on antibody profile.

C  The accuracy of the models is evaluated by making predictions for subjects not used in model training and comparing observed infection rates (blue dashed) to predictions, according to the models based on the two different antibody profile types (Fc Array in pink and titer in black).

the control group animals, leaving a total of 82 features for modeling (Appendix Fig S1C).

**Predictive analysis framework**

In order to characterize the generalizability of our conclusions regarding the utility of using response qualities in addition to response magnitude, we employed a predictive analysis framework in which models were trained using some subjects and then used to make predictions for held-out subjects, thereby enabling evaluation of the accuracy and robustness of the predictions. This approach stands in contrast to statistical measures of correlation over all the subjects, which may suffice to summarize some strong relationships between features and outcomes in a given set of subjects, but do not quantify the generalizability of such relationships to new subjects. The features contributing to a robust, high-performance predictive model can be expected to be reliable, but some features that appear as correlates when characterizing all subjects may be omitted from such a model because the correlation is not sufficiently generalizable or because it does not add predictive value beyond other features included in the model.

Predictive models of protection and adjuvant-specific group differences were built from the titer and Fc Array data, such that the models would predict an animal's risk of infection and adjuvant group based solely on the post-vaccination, pre-challenge antibody profile (Appendix Fig S2A). Protection models employed a multivariate survival analysis based on Cox proportional hazards (CoxPH) regression to predict an animal's risk of infection, while adjuvant group models used a regularized logistic regression approach to predict, or classify, an animal's adjuvant group. For both objectives, modeling was performed separately for the titer features and for the Fc Array features, as well as for a titer-adjusted feature set generated by mathematically rendering the Fc Array features uncorrelated with titer (Appendix Fig S2B). In the case of Fc Array data, feature selection was performed to reduce the risk of overfitting and enhance the interpretability of models derived from this "wide" data with many more features than animals. Prediction accuracy and robustness were evaluated with repeated cross-validation (i.e., training a model on a subset of the samples and making predictions on the remaining samples) and permutation testing (i.e., training "negative control" models on data with the protection and adjuvant group information permuted, or shuffled, with respect to

the antibody features). In order to visualize the features contributing to the predictions, a final model was trained using all animals. The process is summarized in Appendix Fig S2A, and details are provided in Materials and Methods.

## Antibody profiles are predictive of challenge outcome

Multivariate survival analysis was performed using the Fc Array antibody profiles to model risk of infection for the immunized animals and identify antibody features capable of robustly predicting the challenge outcomes. The group-wise risk of infection according to the final model closely mirrored the observed challenge data, with the predicted Kaplan–Meier (KM) curves not significantly different from the observed ones by log-rank test (Fig 2A). Strikingly, the IL-12 group was modeled to have a distinct survival trend from the others, even though the modeling approach relied only on antibody profiles and did not explicitly include group information. Moreover, predictions of risk for individual animals were also accurate, with the animals predicted to be better protected generally withstanding more challenges than those predicted to be at greater risk, as evidenced by a concordance index (C-index) of 0.74 ($P < 10^{-13}$) in a representative cross-validation run (Fig 2B). Overall, the animals in the IL-12-adjuvanted group were predicted to have significantly lower risk of infection compared to the others, again relying just on antibody profiles to explicate implicit group-based differences (Fig 2C). The robustness of the modeling approach was confirmed by repeated cross-validation and permutation testing, with models trained on actual data significantly ($P < 0.05$) and substantially (Cliff's Δ: large) outperforming the models trained on permuted data (Fig 2D). This difference suggests that the predictive performance of the model was due to meaningful relationships between the features and protection rather than by chance, since models trained on data with such relationships disrupted by permutation did not perform nearly so well.

The modeling process identified a number of correlates of protection and risk. In particular, the final model was able to accurately predict the challenge outcome using four such antibody features, of which three were correlated with protection and one with increased risk of infection (Fig 2E). Strikingly, all three protective correlates were antibodies capable of binding the complement cascade initiator C1q. Unexpectedly, these antibody protective correlates had specificity for each of the antigen components in the DNA vaccine (Env, Gag, and Pol). This diversity of antigen specificities was surprising since only FLSC (i.e., the Env antigen) was included in the protein boost, resulting in antibody titers to Env that were several orders of magnitude greater than those for Gag or Pol at the time of challenge. The Env- and Gag-related features were found to have significantly elevated magnitudes in the IL-12-adjuvanted group (the only significantly protected group) as compared to the others (Appendix Fig S3A and B), whereas the Pol-specific feature was elevated among all three adjuvanted groups compared to the non-adjuvanted group (Appendix Fig S3C). In contrast, the correlate of risk, an rhFLSC-specific response mediating FcγR2A binding, was not statistically different between any pair of groups (Appendix Fig S3D). Since Gag and Pol were in the DNA prime, but not the boost, the observed group-level differences are likely directly attributable to DNA priming.

It should be noted that the features contributing to the final model were identified by aggressive feature selection and thus are likely not be the only pertinent correlates. In this regard, other features could likely be incorporated into alternative models capable of similar prediction accuracy. While it would be intractable to assess all possible models, the backward feature elimination process employed here for feature selection iteratively eliminated features contributing little to training set performance and thereby yielded features that are representative of the best correlative trends. In this regard, it is expected that other features could be incorporated into alternative models capable of similar prediction accuracy. Thus, we performed a substitution analysis, replacing each of them one at a time with alternatives. Substitution of other Gag-specific features tended to maintain high accuracy in cross-validation testing (Fig 2F), indicating that there was a broad range of Gag-specific antibody-directed activity that was equally predictive of protection. To further explore the potential importance of Gag-specific features, we determined the extent to which model accuracy was impacted by the exclusion of all Gag-specific features. The resulting "Gag-less" models retained similar accuracy (C-index: $0.75 \pm 0.02$) and robustness ($P < 0.05$; Cliff's Δ: large) as those trained using the complete feature set. A similar exercise was carried out to estimate the impact of elimination of Pol- and Env-specific antigens on the prediction accuracy. While the "Pol-less" models maintained prediction accuracy on par with the original models, the "Env-less" models showed a significant drop in accuracy (Appendix Fig S4). Taken together, these findings suggest that humoral responses targeting Gag and Pol do not contribute to models of protection as much as those targeting Env do.

---

**Figure 2. Protection from challenge can be robustly modeled with antibody profiles.**                                          ▶

A  The predicted survival probabilities in the final model closely match observed KM curves for the IL-12 adjuvanted group (red) and for the others (blue). Log-rank tests indicate insignificant difference between predicted (solid) and observed (dashed) curves. $n = 8$ for IL-12 and $n = 24$ for Others.

B  The predicted relative risk of infection for each animal in the representative eightfold cross-validation run (relative to mean at 0, horizontal dashed line) closely matches observed challenge data (concordance (C)-index). The colors represent the adjuvant groups (groupID) as shown in the central legend box to the right side of panel (D).

C  Animals in the IL-12 adjuvanted group (red) have significantly lower predicted risk of infection than those in the combined other groups (blues) in the representative eightfold cross-validation run (Wilcoxon–Mann–Whitney). $n = 8$ for IL-12 and $n = 24$ for Others. The colors represent the adjuvant groups (groupID) as shown in the central legend box to the right side of panel (D). The horizontal line in the box represents the median, the upper and lower limits represent the 3rd and 1st quartiles respectively, and the whiskers extends from the upper/lower limit to the highest/lowest value that is within 1.5 * (interquartile range) of the limit.

D  The approach is robust, with models trained on actual data consistently obtaining high C-indices (100 repetitions of eightfold cross-validation yield mean $0.73 \pm 0.02$) and significantly (tail probability) and substantially (Cliff's Δ) outperforming those trained on permuted data ($0.61 \pm 0.08$) and baseline C-index for random prediction (0.5).

E  A small set of features (columns) contribute to the final model (coefficients in bars; top panel), with one predictive of risk and three of protection (Cox PH $P$-values: **$P < 0.01$; *$P < 0.05$; -, not significant). The individual animals (rows) are colored by adjuvant group and ordered in ascending order of time-to-infection.

F  Substitution analysis reveals co-correlates of the features from the final model (E), dominated by Gag specificities as well as ability to bind C1q.

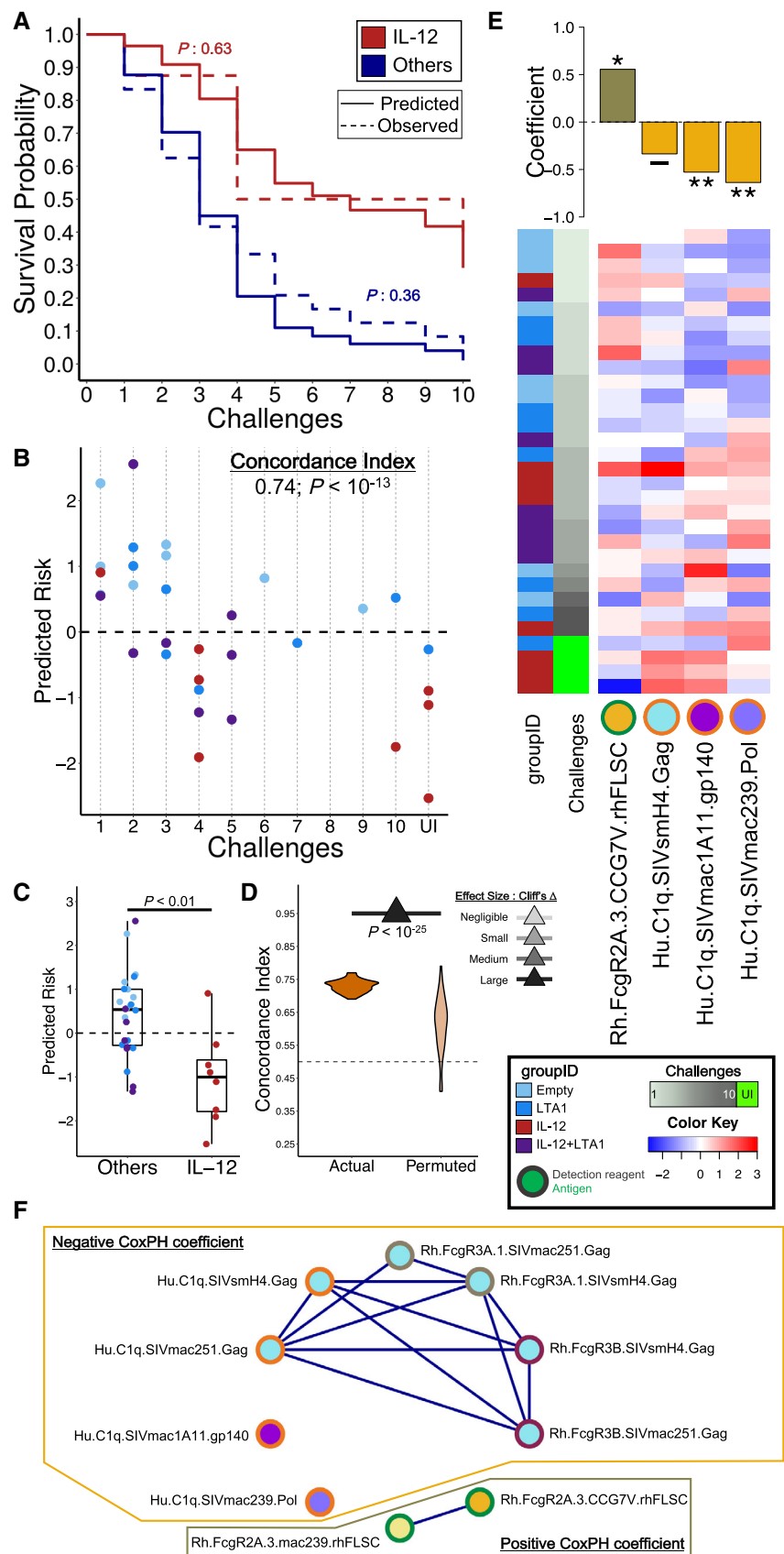

**Figure 2.**

The previously published analysis of this study identified a balance between humoral and cellular immunity as key to protection (Fouts *et al*, 2015). The survival models presented here already performed well using only antibody features, and inclusion of cellular response features in the modeling inputs did not yield better performance. However, the cellular response did prove to be useful in explaining some of the misestimates of risk made in the survival models (Appendix Fig S5). In particular, of the sixteen animals that were predicted to have low risk of infection (risk < 0) according to the representative humoral-based survival model, eight were infected in four or fewer challenges. Of these eight, we found that three had a high cellular response as characterized by IFNγ expression by stimulated T cells (Appendix Fig S5A) and thus appear to have suffered from a lack of immune balance as previously described (Fouts *et al*, 2015). Similar analysis of the other previously described correlate, ADCC, did not provide additional insights (Appendix Fig S5B), presumably because the antibody features contributing to the predicted risk already captured the quality of the humoral response well. With animals stratified according to the combination of predicted risk representing the humoral response and IFNγ representing the cellular response, KM curves support the importance of the previously observed balance between these two components (Appendix Fig S5C, cf. (Fouts *et al*, 2015) Fig 4F).

### Antibody qualities beyond titer are essential for robust prediction of protection

Survival analysis identified a set of correlates focused on Fab domain recognition of Env, Gag, or Pol and Fc domain interactions with C1q, an initiator of the complement cascade, along with substitutable co-correlates. However, it is possible that the main driver of protection (and therefore largest contribution to the model) was simply the magnitude of the response(s). We tested this possibility in two ways, applying the same modeling approach to two different sets of features, termed "titer" and "titer-adjusted". A "titer-adjusted" set of features was also derived from the original Fc Array data by mathematically removing the influence of the corresponding antigen-specific titer features by orthogonal projection (Appendix Fig S2B). This procedure thereby yielded antibody features that were uncorrelated with titer. These titer-adjusted features thus served as a deeper characterization of antibody qualities, in that they measured characteristics of the response beyond magnitude. It should be noted that these titer-adjusted antibody features remained quantitative, using numerical values; the important point is that the resulting profiles were uncorrelated with the titer profiles.

Models using only the titer features (i.e., magnitude of antibody responses in the Fc Array) displayed much poorer performance at predicting the group-level or animal-level risk of infection (Fig 3A–C). Indeed, statistical comparisons from repeated cross-validation showed that models trained on actual titer-only data did not do significantly or substantially (Cliff's Δ: small) better at predicting protection than those trained using randomly permuted titer data (Fig 3G). In contrast, models trained using only the titer-adjusted Fc Array features displayed performance that was as good as that of the original models trained using Fc Array features (Fig 3D–F). The C-indices from repeated cross-validation with titer-adjusted features showed that those models have a predictive performance that is similar to that of Fc Array-based models (Figs 3G and 2D). Though the robustness tests did not show a statistically significant difference (*P*: 0.09), the effect size confirmed a substantial difference (Cliff's Δ: large) between the actual and permuted models (Fig 3G).

### Antibody profiles reveal adjuvant-specific responses distinguishing groups

Since the IL-12-adjuvanted group showed a significantly lower infection rate than the combination of all other groups, we used a regularized binomial logistic regression approach to build models classifying the IL-12-adjuvanted group versus the others based on antibody feature profiles, with and without titer components. The regularized modeling process inherently enabled identification of informative features, here components of the humoral response specific to the IL-12-adjuvanted group. Cross-validated classifiers trained on the original Fc Array features were able to clearly discriminate the two groups (i.e., IL-12 versus all other animals), as was evident from the accuracy of 75% (Fig 4A). Repeated cross-validation and permutation testing indicated that the classification approach was robust, showing both a statistical (*P* < 0.05) and substantial (Cliff's Δ: large) difference in performance between models trained on actual and permuted data (Fig 4D). For inspection, a final classification model was trained using all animals relying on a small set of features to discriminate the IL-12 group from the others (Fig 4B and C). Of the six Fc Array measurements employed by the model, four were associated with the IL-12 group, two of which (corresponding to the ability of Env- and Gag-specific antibodies to interact with C1q) had also contributed to the protection models as correlates of protection, while two others (corresponding to the ability of Env-specific antibodies to interact with FcγR3A) were indicative of a response common within the IL-12 group, but not necessarily of good protection (Appendix Fig S6).

---

**Figure 3. Robust protection modeling depends on antibody qualities beyond titer.**

A–F   Models were trained using either (A–C) titer features or (D–F) titer-adjusted features. (A & D) Observed KM curves and predicted survival probabilities. The predicted (solid) curves for the model using titer features (A) are significantly different from the observed (dashed) ones, while those for the model using titer-adjusted features (D) are not (log-rank test). (B & E) Observed time-to-infection versus predicted risk of infection according to representative eightfold cross-validation runs. Predictions from the titer-adjusted model (E) are much more concordant with observation (C-index) than those from the titer model (B). (C & F) Group-wise differences in predicted risk of infection from the representative eightfold cross-validation runs. The titer model (C) does not predict the observed difference in protection between groups (Wilcoxon–Mann–Whitney), while the titer-adjusted one does (F). *n* = 8 for IL-12 and *n* = 24 for Others.

G   C-indices from repeated cross-validation and permutation testing, using the three different sets of features. Titer-adjusted data maintain substantial (Cliff's Δ) difference between using the actual data and the permuted data. There are also significant (tail probability) and substantial (Cliff's Δ) differences between titer-only cross-validation results and others, but not between the original Fc Array data and titer-adjusted data. The pair-wise comparison between actual models using three feature sets was done by measuring the tail probability of the mean of one distribution with respect to the other. The horizontal lines represent the mean C-index for each data type and for random prediction (0.5). One hundred repetitions of 8-fold cross-validation.

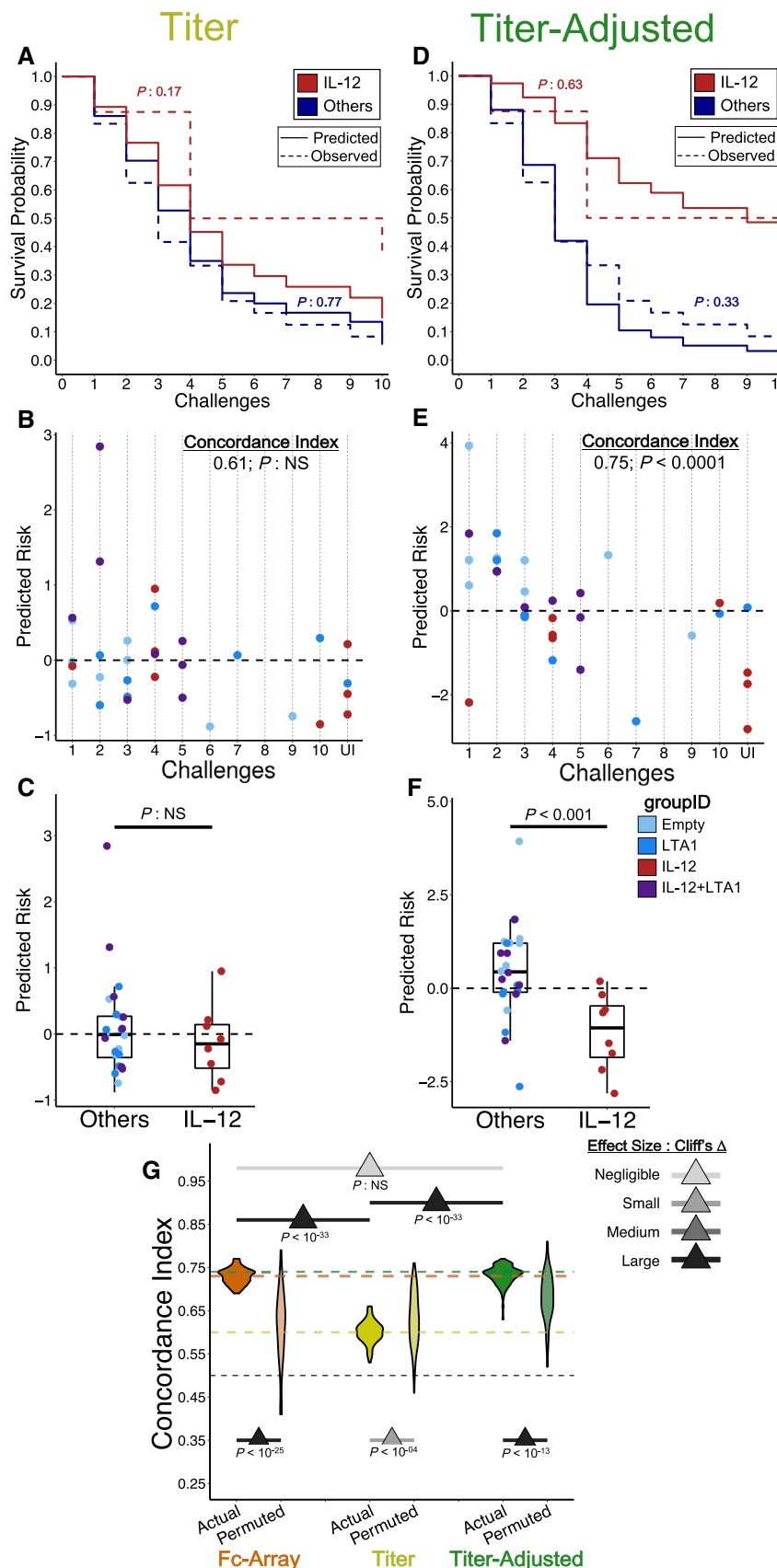

Figure 3.

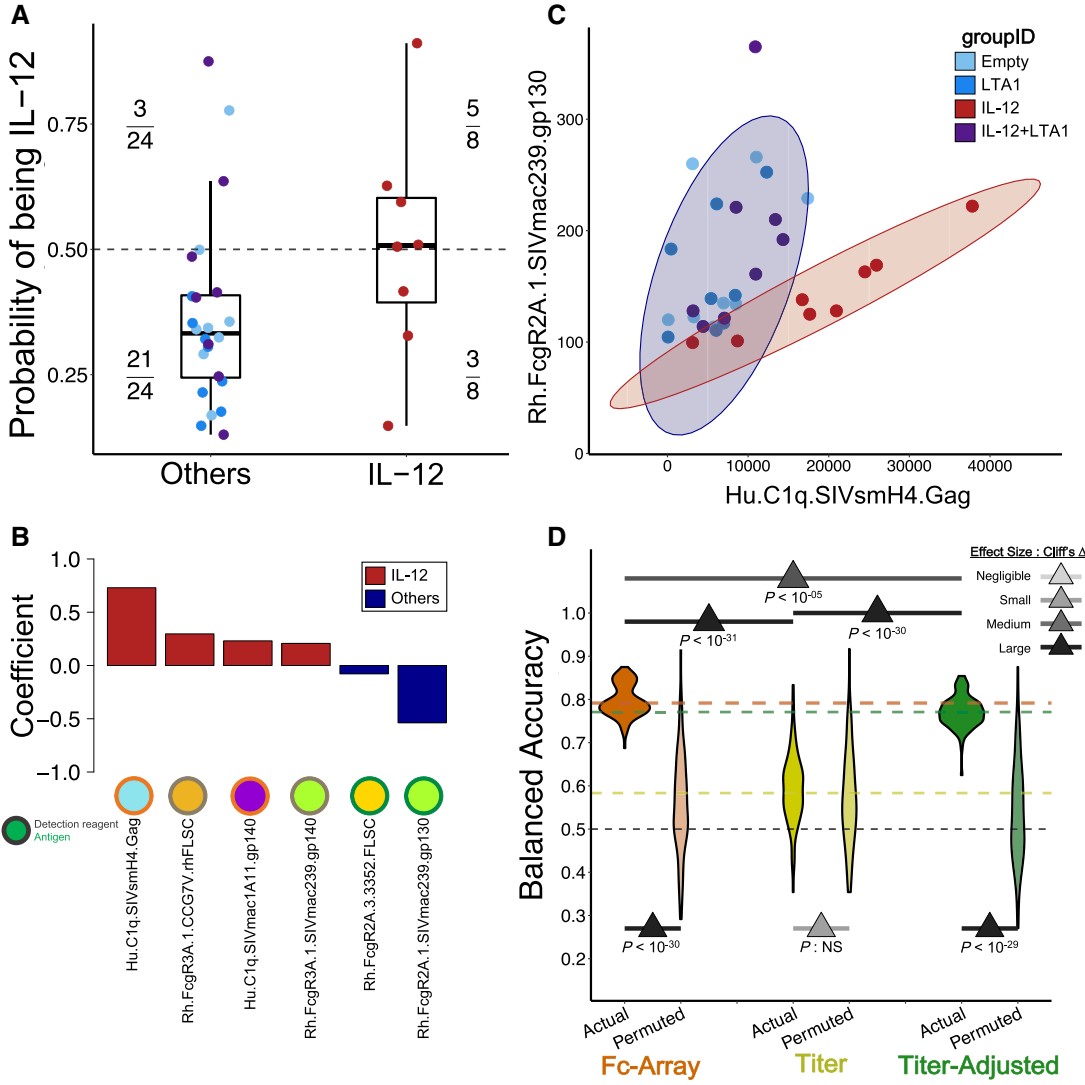

**Figure 4.  Antibody profiles distinguish IL-12 adjuvant-specific responses from others.**

A   Predicted probability of being in the IL-12 group according to the binomial logistic classifier in the representative eightfold cross-validation run. Three animals (points, colored by group) from the combined other adjuvant groups are predicted to be members of the IL-12 group (above the 0.5 decision boundary, dashed line), and likewise 3 out of 8 IL-12 animals are misclassified as belonging to the "others" group. $n = 8$ for IL-12 and $n = 24$ for Others.

B   The final model classified animals into groups based on a linear combination of a small set of features associated with the IL-12 group (positive coefficients, red) vs. others (negative, blue).

C   Two of the features in the final model clearly separate the other groups.

D   Classifiers obtain robust performance using the original Fc Array features or the titer-adjusted features, but not the titer-only features, as revealed by tests of significance (tail probabilities) and magnitude (Cliff's Δ) in results from repeated cross-validation and from permutation. The pair-wise comparison between actual models using three feature sets was done by measuring the tail probability of the mean of one distribution with respect to the other. One hundred repetitions of 8-fold cross-validation.

A comparison of IL-12 group classification performance with models trained on titer features and titer-adjusted features showed that, as with survival analysis, features capturing antibody Fc binding properties and not just titer were again necessary to robustly identify adjuvant-associated attributes of the humoral response (Fig 4D). Again, cross-validated models trained using only titer features performed significantly ($P < 0.01$) and substantially (Cliff's Δ: large) worse than those trained on the original Fc Array features. In fact, the permutation tests indicated that titer features performed no better

in distinguishing the true classes than in distinguishing permuted classes (Fig 4D, center). On the other hand, the titer-adjusted features (Fig 4D, right) performed comparably to the original Fc array features (Fig 4D, left; see also Fig 2D). Thus, this analysis shows that antibody titer alone was not sufficient to identify adjuvant-associated effects on the humoral response in this immunization study.

Up to this point, we have focused on distinguishing the best protected group, IL-12, from the others, but high-resolution data offer the chance to try to further distinguish among the other

groups. As such, a multinomial logistic regression approach was used to classify which of the four adjuvant groups each animal belonged to, based solely on feature profiles from the Fc Array. The overall methodology followed that established for two-way classification (IL-12 group versus all others), but using a multinomial model instead of a binomial one in order to simultaneously classify all groups. The resulting model's accuracy was around 58% (compared to the 25% baseline expected for a 4-group classification), with animals from the LTA1 and LTA1 + IL-12 groups being the most commonly confused (Appendix Fig S7A and B). The repeated cross-validation performance was still robust compared to the performance of models trained on permuted data (P < 0.01 and Cliff's Δ: large) (Appendix Fig S7C). The features identified in the final multi-way classification model as being associated with the IL-12 group were also all identified in the previous two-way classification model, albeit with different relative magnitudes of the coefficients (Appendix Fig S7D). The top two features (by coefficient magnitude) associated with the IL-12 and Empty (i.e., no DNA adjuvant) groups clearly separated them from the other groups (Appendix Fig S7E and F). It should be noted that the top two features for the Empty group were negatively associated with the group indicating a reduced response compared to the other three groups (Appendix Fig S7E).

## Quality-based models outperform titer-based models in a distinct SIV vaccine study

We next sought to evaluate the generality of the observation that humoral response quality, beyond magnitude alone, is important in analyzing protection. Recently, Fc Array and functional measurements were collected for animals in a distinct SIV vaccine study

(employing a DNA prime-Ad5 boost vaccine regimen), and the analysis pointed to strikingly different antibody properties and correlates of protection associated with different routes of vaccination (Ackerman et al, 2018). In that study, animals vaccinated with a SIVmac239 Env immunogen (administered either intramuscularly or mucosally) exhibited significantly better protection against infection compared to those vaccinated with a mosaic Env immunogen. Predictive survival analyses found that protection achieved via the two routes of vaccination with SIVmac239 Env was associated with two different antibody-mediated functional responses, monocyte and neutrophil phagocytosis, and the FcgR2a and C1q binding capacity of envelope- and variable loop peptide-specific antibodies.

The Fc Array data from that study provided an excellent opportunity to further evaluate the predictive capacity of models trained on IgG response magnitudes alone or those that also incorporated other aspects of antibody qualities. We thus applied the same predictive analysis framework to the data from that study, building and evaluating survival models of protection and classification models of vaccine group (three groups: intramuscular and aerosol based on a SIVmac239 antigen, as well as intramuscular based on a mosaic antigen). Comparisons of survival models revealed that those using Fc Array or titer-adjusted Fc Array data performed significantly and substantially better than those based on titer data alone, with essentially random performance observed for titer-only models (Fig 5A). For vaccine group classification, all three feature sets yielded models significantly better than random, with Fc Array models significantly and substantially better than the other two (Fig 5B). Since titer plays a considerable role in distinguishing groups here, it is important to maintain it in the models. We see that once again it is the combination of quality and quantity that yielded the best models.

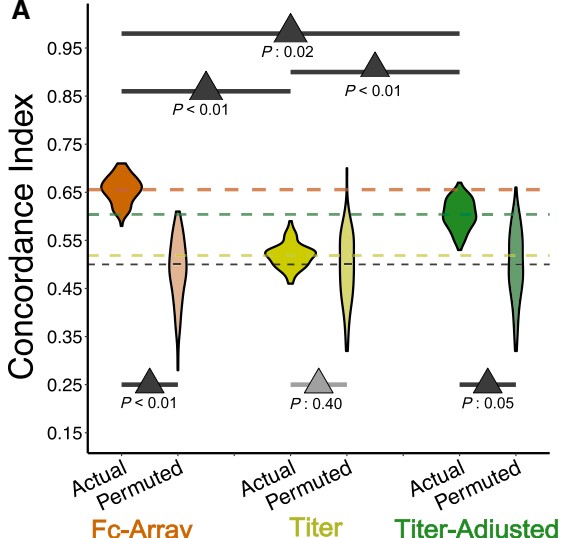
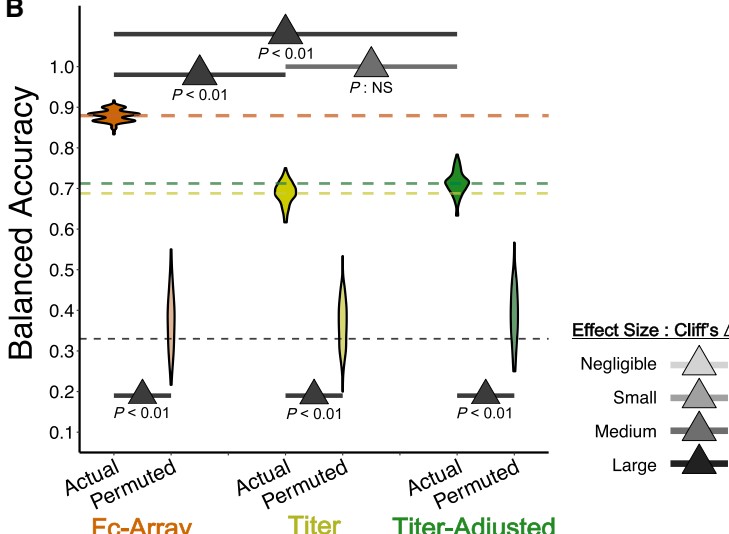

**Figure 5. Models incorporating antibody quality data outperform those using titer alone in a distinct vaccine study (Ackerman et al, 2018).**

A  C-indices from repeated cross-validation and permutation testing, using the three different sets of features. The original and titer-adjusted Fc Array data maintain substantial (Cliff's Δ) differences between using the actual data and the permuted data. Their performance is also significantly (tail probability) and substantially (Cliff's Δ) better than that for titer-only models. The horizontal lines represent the mean C-index for each feature set and for random prediction (0.5).

B  Classifiers are all better than random (comparing models with permuted data), but those with Fc Array data are significantly (tail probability) and substantially (Cliff's Δ) better than those with titer-only and titer-adjusted data, which show a significant drop in performance.

## Discussion

Systems approaches enable high-resolution characterization of key factors associated with robust immune responses. Here, the high-throughput Fc Array was used to dissect humoral responses to vaccination in an SIV challenge study, revealing antibody properties that were associated with protection as well as properties that were different between vaccine groups. While the use of multi-dimensional, "wide" data raises the risk of focusing on artifacts of a given training set, the consistent results here from the models across multiple cross-validation runs, and in contrast to models using shuffled data, provide evidence of their generality, accurately and robustly predicting results for unseen samples and identifying features with potential mechanistic relevance. Survival and classification analyses demonstrated that characterizing antibody Fab and Fc properties played a much more critical role than measuring antibody titers alone. Antibody feature-based models, even with titer components mathematically removed, consistently and robustly predicted protection and distinguished between adjuvant groups, whereas titer-only based models yielded essentially random performance.

This result should not be interpreted broadly to suggest that the titer, or magnitude of the humoral response, is not an important factor in modeling or achieving protection from infection. Instead, this work provides a framework and methodology by which to assess the relative influence of qualitative and quantitative variability in the humoral response and evaluate the extent to which particular antibody properties can enable better characterization of responses. In cases where very divergent antibody types are elicited, expected relationships between response magnitude and challenge outcomes can be obscured. Importantly, in studies where different vaccine candidates elicit similar quantities of antibodies, segregating the response by antibody features can help factor those vaccine-specific differences and reveal their relationships to protection. Therefore, it can be highly beneficial to characterize and investigate both quantity and quality of the response to identify correlates of protection in vaccine studies.

Here, protection was modeled with a multi-dimensional, multi-group CoxPH survival model, combining animals from all adjuvant groups and employing aggressive feature filtering, to incorporate only a small number of features. The use of a Cox modeling approach follows that used in recent studies (Bradley *et al*, 2017; Ackerman *et al*, 2018), so as to avoid having to discretize protection levels (Chung *et al*, 2015; Vaccari *et al*, 2016). We found that to avoid overfitting, it was necessary to perform feature pre-filtering and employ greedy feature elimination methods to narrow down the number of features used by the Cox model (Frejno *et al*, 2017); regularization techniques (Simon *et al*, 2011) displayed substantially poorer performance, resulting in empty models (i.e., with no features selected), likely due to the semi-parametric formulation of the Cox PH model and the small sample size of the study. However, we note that the greedy feature selection framework does not guarantee finding the "best" features and in addition could be computationally expensive in studies with larger feature sets. Recently, other survival analysis methods, which model protection as a complex non-linear function of the input features, with potentially better prediction accuracy than the linear models, have been proposed (Ishwaran *et al*, 2014). However, those methods did not perform any better than linear models in this study, and hence, Cox PH was favored for ease of interpretability. A final limitation of the approach here is the use of a single model for animals from all groups instead of group-specific models. This was required in order to obtain a sufficient sample size for the model; interestingly, the model reflected the group-level differences in its predictions even without explicitly encoding group identification. If enough animals were available to sufficiently power the analysis, group-specific models could complement the unified model by factoring out major differences between groups and thereby potentially enable identification of additional correlates of protection associated with the individual groups.

The predictive analyses in this work have enabled a deep exploration of the effects of different adjuvants on the humoral responses induced by the SIV antigens used here. In general, the predictive modeling approach allows moving beyond characterizing high-level differences in efficacy due to differences in adjuvants, providing finer-grained insights into how responses varied consequently. Here, the features identified by the group classifiers show that the IL-12-adjuvanted group exhibited a stronger response for Gag- and Env-specific antibodies binding complement (C1q). It is interesting that the same two features were also identified by the survival models as correlates of protection across all the groups. These two results indicate that the IL-12-adjuvanted group exhibited a distinct response compared to the other three groups, which also resulted in better protection against SIV challenge as compared to the others. Further, while the substitution analysis for survival models showed that there was a broad range of antibody responses against the Gag protein that correlated with protection, models restricted to exclude these features performed equally well, suggesting that Gag-specific features did not play a crucial role in models predicting risk. Alternatively, it is possible that the antibody responses to Gag that correlated with protection could be a surrogate for another antibody response or mechanism that directly contributed to protection. Nonetheless, these findings reinforce the observation from previous studies (Schadeck *et al*, 2006; Chong *et al*, 2007; Robinson *et al*, 2007; Jalah *et al*, 2012) that the IL-12 adjuvant not only elicits an enhanced Gag-specific immune response, but also elicits a broad protective response. Here, inclusion of the IL-12 adjuvant has clearly helped provide better protection in this DNA prime, protein boost regimen. Although studies in humans have reported that antibodies against p24 Gag antigen are elevated in HIV-1 controllers compared to the chronic progressors (Banerjee *et al*, 2010; French *et al*, 2013), it is not known whether these anti-Gag humoral responses directly contribute to viral suppression. Similarly, one of the strongest features from our models that predicted protection was Pol-specific antibodies binding C1q. However, this feature was not unique to the IL-12 group. Follow-up studies would be required to assess the intriguing possibility that the Gag- and/or Pol-specific correlates of protection observed here directly contribute to protection or are a surrogate for another response that is directly responsible for protection. Our observation linking virus-specific antibodies that interact with C1q to protection suggests the potential importance of complement-dependent anti-viral activities in vaccine-mediated protection. To this end, it is intriguing to note that in the RV144 study, polyclonal sera from uninfected vaccine recipients demonstrated elevated complement activation as compared to that from infected vaccinees (Perez *et al*, 2017). However, follow-up

studies would be required to definitively link any of the observed correlates to mechanisms of protection, and it is important to note that the ability of different sets and combinations of features to accurately model infection outcomes points toward the limitations of equating models with mechanisms, particularly in a small-sample study.

In summary, this experimental and analytical approach offers a path forward in correlates analysis when the quality of the antibody response is sufficiently diverse among individuals or between regimens so as to mask or eliminate relationships between the magnitude of response and degree of protection. Although the analysis of this adjuvant study found antibody quality to be more predictive of protection than antibody quantity, the result does not discredit the general use of antibody titer in providing correlates of protection; possessing high-quality antibodies alone would not be sufficient to render protection without an adequate quantity of antibody present. However, this work suggests that even when antibody titer is a correlate of vaccine efficacy, and especially when it is not, studies of antibody properties can yield deeper insights into how the immune system responds. Ultimately, predictive modeling of antibody qualities can provide detailed correlates with robust predictive power, suggest directions for further vaccine improvement, and enable the discovery of potentially mechanistic correspondences across studies.

# Materials and Methods

### Multiplexed IgG titering and Fc Array

To characterize humoral responses in a high-throughput manner, the Fc and Fab characteristics of circulating antibodies were simultaneously probed using the previously described Fc Array (Brown *et al*, 2012, 2017). The assay utilized an on-bead, antigen-specific purification step to capture circulating antibodies. Each unique antigen was coupled to a fluorescently coded bead, thus allowing the FlexMap3D instrument to distinguish antibody specificity in a multiplexed assay. After incubating the beads with the test samples, unbound antibodies were removed and the Fc regions were probed using phycoerythrin-conjugated detection reagents including human and rhesus Fc$\gamma$Rs, human C1q, MBL, and anti-IgG (listed in Appendix Fig S1). Samples were subsequently analyzed on a FlexMap3D instrument (Luminex) which reported antigen-specific, median fluorescence intensities (MFI) for each of the detection reagents. Fc Array measurements were pre-processed for quality by eliminating measurements that did not show a significant (Wilcoxon–Mann–Whitney, $P < 0.01$) difference between the unvaccinated control and any vaccinated group. Features for which the Fc detection reagent was IgG, measured at two serum concentrations, were considered to be "titer" features (Appendix Fig S1C) and were held out from the main analysis on Fc Array measurements (Appendix Fig S1B).

### Survival analysis

Models were trained to predict the risk of infection at each challenge point using a previously described multivariate survival analysis approach (Bradley *et al*, 2017) based on Cox proportional hazards (CoxPH) regression (Cox, 1972). In summary, the method employs the following components:

*Feature pre-filtering:* To reduce the risk of overfitting by CoxPH due to "wide" data (Vinzamuri & Reddy, 2013; Laimighofer *et al*, 2016), only the top 10% of features correlated with protection by polyserial correlation coefficient (Drasgow, 2004) were considered, and a non-redundant subset was selected so as to eliminate correlated features (Pearson correlation coefficient > 0.8).

*Model training and feature selection:* Models were trained and tested in an eightfold cross-validation setting with animals randomly split into folds using stratified sampling to ensure that animals from each group were included in each test set. For a training set, CoxPH models were trained using the R package "survival" (Therneau & Grambsch, 2000). Greedy backward feature elimination (Guyon & Elisseeff, 2003) was performed to iteratively reduce the set of features contributing to models and thereby further reduce the risk of overfitting. Features were eliminated as long as training likelihood was no more than 25% worse than the initial model using the pre-filtered features.

*Risk prediction and performance evaluation:* CoxPH models were used to predict each animal's risk of infection relative to the mean risk over all animals. These relative risk predictions enabled assessment of a model's predictive performance versus observed time-to-infection, according to the concordance index (C-index) metric (Harrell *et al*, 1996). In order to estimate the variation in performance, cross-validation was repeated 100 times with different splits of animals. In order to provide "negative control" models trained on incoherent data with the same characteristics as the real data, a permutation testing approach (Ojala & Garriga, 2010) was employed, where permuted data were generated by randomly shuffling the challenge labels and then following the same exact evaluation process to filter features and train and test models. Performance differences between models using actual versus permuted data were characterized by (i) determining the mean C-index of models using actual data and assessing the tail probability of this value with respect to the distribution of C-indices of models using permuted data, and (ii) the magnitude of the differences between the C-index distributions of models using actual data versus those using permuted data (effect size by Cliff's $\Delta$).

*Representative models:* For inspection, a representative run of eightfold cross-validation was performed using features that contributed to at least 90% of the models obtained over the repeated cross-validation. The representative cross-validation run enabled plotting of each animal's predicted risk when it was part of the testing set. Furthermore, to evaluate group-wise risk and plot aggregate KM curves, a final model was trained using all the animals and the same set of high-frequency features. The survival probabilities for each adjuvant group were estimated by making predictions on "mean" animals (i.e., with feature values taken as the mean values over their respective groups).

*Substitution analysis:* To identify co-correlates, features correlated to each feature in the final model (Pearson correlation coefficient > 0.75) were considered as possible substitutes. For each such possible substitute, an eightfold cross-validation was performed to assess performance of a variant of the final model using the substituted instead of the corresponding final feature.

## Titer-adjusted features

A set of features linearly independent of titer was developed by means of linear orthogonal projection. Let $c_a$ be a mean-centered Fc Array feature vector, over the subjects, for antigen $a$ and some Fc property. Let $g_a$ be the mean-centered, unit-length IgG feature vector, averaged over low and high concentrations, for the same subjects and the same antigen $a$. Compute $\hat{c}_a$ as the orthogonal projection of $c_a$ onto $g_a$ (Eqn 1). Then, $\vec{c}_a^{\perp}$ (Eqn 2) is independent of (orthogonal to) the titer feature $g_a$ and is included in the titer-adjusted feature set. The resulting titer-adjusted feature set consists of Fc Array features that are linearly independent of antibody titer (i.e., they are uncorrelated with titer measurements specific to the respective antigen).

$$\hat{c}_a = \frac{\vec{c}_a \cdot \vec{g}_a}{\vec{g}_a \cdot \vec{g}_a} \vec{g}_a \tag{1}$$

$$\vec{c}_a^{\perp} = \vec{c}_a - \hat{c}_a \tag{2}$$

## Classification of adjuvant groups

Least absolute shrinkage and selection operator (LASSO)-regularized binomial logistic regression (Cox, 1958) was used to develop models that linearly combine relatively sparse sets of features in order to distinguish animals in the IL-12 group versus the other three groups. Model training was performed via the R package "glmnet" (Friedman *et al*, 2010) with default options, and the penalty parameter (lambda) for regularization that achieved the lowest classification error was used to train the final model.

Modeling performance was assessed in terms of the balanced accuracy (mean true positive rate) over 100 repetitions of eightfold cross-validation. Training/testing splits were constructed so as to ensure that each testing set included at least one animal from each group. Robustness was evaluated with permutation testing, namely, repeating the same process with randomly shuffled group labels. Performance differences between models using actual versus permuted data were again characterized as was done for survival analysis, computing the tail probability and effect size. Visual inspection of predicted classes was based on a single run of eightfold cross-validation, while visual inspection of regression coefficients was based on a final model trained using all the animals.

For classification distinguishing all four adjuvant groups, the same approach was employed but using glmnet's multinomial logistic regression model (Friedman *et al*, 2010) instead of its binomial one.

## Data availability

All data and R code to perform the described survival analysis, group classification, and titer adjustment are available on Zenodo at https://doi.org/10.5281/zenodo.2614007.

**Expanded View** for this article is available online.

## Acknowledgements

These studies were supported by the Bill and Melinda Gates Foundation (OPP1032817, OPP1146996, and OPP1114729) and the HHS|National Institutes of Health (NIH) National Institute of Allergy and Infectious Disease and National Institute of General Medical Sciences (R37 AI080289, R01 AI102291, R01 AI131975, P01 AI120756, R01 AI102660, R44 AI074334, R44 AI102702, and R44 AI091567), and HHSN272201100016C. We thank Judith T. Lucas and David M. Beaumont for binding antibody assays, and Robin Flinko and Nicole L. Yates for technical support. The following reagents were obtained through the AIDS Reagent Program, Division of AIDS, NIAID, NIH: SIVmac251 BK28 pr55 Gag Recombinant Protein from the NIAID Vaccine Research and Development Branch, Division of AIDS, NIAID; SIVmac1A11 gp140 Recombinant Protein from the Division of AIDS, NIAID (Spira *et al*, 1996); SIVsmH4 p55 Gag from the Division of AIDS, NIAID; and SIVmac239 gp130 from the Division of AIDS, NIAID (Hill *et al*, 1997).

## Author contributions

SP performed the data analysis. CB-K, MEA, TRF, MR, GA, GDT, DCM, GKL, GF, JAS, and KB supervised experimental and statistical analysis. CCL, CB, XS, RP, AO-S, RX, WZ, IJP, JAW, and EPB performed assays and aggregated data. SP, KB, JAS, MEA, and CB-K wrote and edited the manuscript.

## Conflict of interest

The authors declare that they have no conflict of interest.

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
