## [Review Process File · Molecular Systems Biology]

Antibody Fab-Fc Properties Outperform Titer in Predictive Models of SIV Vaccine-Induced Protection

Srivamshi Pittala, Kenneth Bagley, Jennifer A. Schwartz, Eric P. Brown, Joshua A. Weiner, Ilia J. Prado, Wenlei Zhang, Rong Xu, Ayuko Ota-Setlik, Ranajit Pal, Xiaoying Shen, Charles Beck, Guido Ferrari, George K. Lewis, Celia C. LaBranche, David C. Montefiori, Georgia D. Tomaras, Galit Alter, Mario Roederer, Timothy R. Fouts, Margaret E. Ackerman and Chris Bailey-Kellogg

Review timeline:

Submission date:	16 th December 2018
Editorial Decision:	4 th March 2019
Revision received:	1 st April 2019
Accepted:	4 th April 2019

Editor: Maria Polychronidou

Transaction Report:

1st Editorial Decision

4th March 2019

Thank you again for submitting your work to Molecular Systems Biology. I would like to apologise once again for the delay in sending you a decision on your manuscript. As I already mentioned in my message earlier today, we had initially secured two reviewers but unfortunately reviewer #1 never returned a report despite repeated reminders. In order to perform an informed evaluation without relying on the single opinion of reviewer #2, we had to invite new reviewers and this considerably delayed the process. An additional reviewer (#3) accepted to evaluate the study and returned their report this past weekend. As you will see below, the two reviewers are overall positive. However, they raise a series of concerns, which we would ask you to address in a revision.

Overall, the reviewers' recommendations are clear. Therefore I think that there is no need to repeat any of the comments listed below. Please feel free to contact me in case you would like to discuss in further detail any of the issues raised by the reviewers.

REFeree REPORTS.

Reviewer #2:

This is an interesting paper where Pittala et al. used a multiplex assay to characterize Fab and Fc function of antibodies elicited by different vaccine regimens described in earlier publications. In the first vaccine study they uses data from an Fc array that included 12 Fab specificity and 10 Fc proprieties and developed a predictive analysis framework to investigate correlate of risk against SIVmac251 infection in vaccinated macaques.

They found that anti envelope abs able to bind the complement cascade initiator C1q was a predictor of a decrease risk of virus acquisition. Analyses that included the titers of antibodies did not do better than the Fc array alone in predicting risk.

Similarly in the second study the Fc array alone scored better than the titers or the combination of both.

Overall the paper demonstrated a contribution of the quality of vaccine-induced antibody response in predicting the risk of virus acquisition. The notion that Systems Serology can be used to monitor immune responses in vaccinees, is very exciting.

What remains unclear and needs to be clarified/addressed for the reader includes :

First Study:

-Was the predictive result of vaccine efficacy obtained with more than one env immunogens ? Did the Fc array on C1q for the rhFLSC immunogen correlate?

- The ADCC titers correlated with the risk of both SHIV and SIV acquisition.

There should be more discussion on why this response was not scored in the predictive analysis framework?

Second study:

This part of the paper is not described in sufficient details and not discussed in the context of the correlates found in the prior work (monocyte mediated phagocytosis and neutrophil mediated phagocytosis)

What Fab specificities correlated in the second study?

Reviewer #3:

In the manuscript by Pittala et al., authors reconsidered the previously published vaccine study in rhesus macaque model of HIV infection [Fouts et al., PNAS 2015]. Fouts et al. have shown that a balance between protective antibody response (ADCC specific for CD4-induced epitopes) and T-cell activation determines the level of protection with low amount of vaccine-generated T cell immunity being critical to high level of protection. The manuscript by Pittala et al. aimed to additionally dissect the humoral response. Protection data from heterologous challenge Study 3 in Fouts et al. were used together with serum data (collected for a single pre-challenge timepoint and analyzed by multiplexed Fc Array) to develop predictive statistical models. Multivariate survival analysis predicted a subject's risk of infection, and regularized logistic regression approach was used to classify adjuvant-specific group differences. The models revealed four antibody correlates of protection with three protective features all pointed to antibodies capable of binding the complement cascade initiator C1q and one increased risk of infection feature involving FcγR2A binding. The models suggested the Fc properties of antibody may be more predictive than the titer, while in some settings the titer still may play a considerable role as has been shown by additional analysis of vaccine study from Ackerman et al. [Nat.Med. 2018].

Major points:

The title of the paper is "Antibody Fab-Fc properties outperform titer in predictive models...", but the "titer" used in the analysis is not clearly defined. The manuscript states -- "Antibody quantity was also assessed at two different serum concentrations by an anti-IgG detection reagent". How do these two measured "titer" features relate to the titer from standard assays like ELISA or neutralization? What do the "high" and "low" features mean? Usually, the titer is defined by the assay and has just one measurement. So, here without explanation it is very confusing.

Authors named the titer feature as "quantitative", but with binary output of "high" and "low" it seems to be still rather qualitative or semi-quantitative.

The overall separation of the Detection Reagents into "qualitative" and "quantitative" is very confusing, because all Fc Array data were obtained, analyzed, and presented in the same manner, and all columns in Figures S1C,D have quantitative content.

The authors based their analysis on Fc Array data collected for a single timepoint. The manuscript says it is "post-vaccination, pre-challenge", but it would be useful if they can specify when after the boost the samples were collected. With 2 weeks between the boost and challenge we may expect better identification of correlates of protection if samples were acquired just before the challenge and not right after the boost. If samples from different animals were taken at different time points within the two week-window, will it be taken into account by the models?

Models based on humoral response alone fail to predict the protection outcome without addition of T cell response in some animals (Figure S5A). It is curious to see a discussion on which of the 4 humoral predictive features and to what extent were involved in these cases.

Very intriguing result of the manuscript is a high role of antibodies able to bind C1q in protection. Could it be indirectly linked to amount of generated T cell response? Did authors try to see the correlations between the three C1q features and IFN γ level? C1q has been reported to limit dendritic cell differentiation and activation by engaging LAIR-1 [Son et al. 2012], so we might expect some link.

Fouts et al. study have shown that addition of second adjuvant LTA1 to IL-12 reduced the high protection observed with IL-12 alone. It would be interesting to see a discussion if any of the specific analyzed Fc features or their combination may account for this effect.

Minor points:

The current description of experimental study [Ref.43] in Introduction is confusing due to numbering 1)-3). I suggest to remove 3) and talk about adjuvant in a separate sentence. Also, the Reader will benefit from overall more accurate description of the experiments here. For example, mentioning the route of prime vaccination and that the boost was with rhFLSC variant different from included in the prime vaccination.

suggestion for Table in Figure 1A to have just one unified description and not four same repeats in columns 3 and 4

Fig.S5 description in the text "three of the six poorly protected animals that were predicted to have low risk of infection". suggestion to highlight those 6 animal points in different color to make it easy for Reader to follow.

Fig.S5 A,C have "!" instead of "gamma"

1st Revision - authors' response

1st April 2019

Overall

Before addressing the individual comments, we thought it would be helpful to answer a general form of a concern that we think might underlie questions from both reviewers: "A particular feature was correlated with challenge, so why did it not show up in the predictive models?"

In this work, we focus on identifying correlates of protection that are predictive, in that a model built using these correlates can make predictions regarding animals that were not used in training that model, and that the accuracy and robustness of these predictions can be quantified. We did so by means of a cross-validation framework, in which models are trained using some of the subjects and predictions are made on the held-out remainder, cycling through different held-out subsets of subjects so that predictions are made for all subjects based on models from some of the other subjects. This predictive testing is more stringent than evaluating correlations over all subjects, as correlations may describe trends that are observable in the data, but may be driven by a relatively small group of the subjects and may not be robustly predictive regarding new subjects. Hence, a predictive model may omit features that it considers likely to generalize poorly, perhaps due to weak overall predictive performance. Furthermore, in order to help both generalizability and interpretability, we used a training approach that sought "sparse" models, using only a small set of features. Consequently, when multiple features had similar trends, typically only one would be used and the others omitted.

We thank the reviewers for raising this important question by way of their comments. We have now included some relevant discussion at the start of the "Predictive analysis framework" section (lines 164-175).

Reviewer #2

What remains unclear and needs to be clarified/addressed for the reader includes :

First Study:

- Was the predictive result of vaccine efficacy obtained with more than one env immunogens ?
- Did the Fc array on C1q for the rhFLSC immunogen correlate?

Authors' response: We found only one Env immunogen (SIVmac1A11.gp140 in Fig 1E) contributed to the most predictive models. Furthermore, we found in our substitution analysis (Fig 1F) no additional Env immunogen contributed to these models.

We measured C1q against three rhFLSC immunogens, and these were weakly correlated with risk of infection as shown in the table below. These weak correlations did not generalize well to a predictive setting.

	Fc Array Feature Name	Concordance Index (p-value)
1	C1q.SIVmac239.rhFLSC	0.60 (0.11)
2	C1q.3352.rhFLSC	0.57 (0.28)
3	C1q.CCG7V.rhFLSC	0.52 (0.65)
	Fc Array Final Model	0.74 (<0.0001)

- The ADCC titers correlated with the risk of both SHIV and SIV acquisition. There should be more discussion on why this response was not scored in the predictive analysis framework?

Authors' response: This paper focused on predictive modeling based on biophysical properties of antibodies, characterizing the extent to which such properties enabled better predictions than magnitude alone. However, since, as the reviewer references, the previous analysis found a correlate of protection involving ADCC titers jointly with the cellular response, we did also study the relationship between antibody-property features and ADCC (Appendix Figure S5C). However, when we included ADCC in our predictive analysis, it was not selected by the models (see preamble discussion re correlation vs. predictive modeling), and it was only weakly correlated with the challenge outcome (Concordance index: 0.49). We note that the previous study of this genetically adjuvanted vaccine ("study 3" in Fouts et al., PNAS, 2015) found ADCC to be associated with reduced infection only when considered jointly with the cellular response, but not on its own, and only in looking at groups that were differentially protected, not in terms of a correlative relationship between these activity and the challenge outcomes observed for individual subjects.

Second study:

This part of the paper is not described in sufficient details and not discussed in the context of the correlates found in the prior work (monocyte mediated phagocytosis and neutrophil mediated phagocytosis)

What Fab specificities correlated in the second study?

Authors' response: We apologize for the insufficient characterization for the second study. We have now added details describing the vaccine regimen, along with functional and Fc array measurements correlated with protection/risk (start of "Quality-based models outperform titer-based models in a distinct SIV vaccine study", lines 358-369). We also note that the motivation to revisit that study in this manuscript was to verify the generality of the observations we made for the first study regarding the importance of antibody properties, and hence only used Fc array data for modeling.

When a survival analysis was performed on the Fc Array data from that study, the prediction model identified a combination of four features listed in the table below. Of the three that correlated with protection, two corresponded to the ability of V1a (variable loop) and G49 (V1b) peptide-specific

antibodies to bind to FcγR2A.4, and one corresponded to the ability of SIV_{mac239}gp140-specific antibodies to bind to the complement cascade initiating C1q protein.

These correlates are addressed in detail in the original publication of modeling for this study (Ackerman et al, Nat Med 2018), and we have added a summary to the present manuscript (paragraph referenced above).

	Fc Array Feature Name	Associated with
1	Hu.FcγR2A.4.high.V1a	Protection
2	Hu.FcγR2A.4.high.G49	Protection
3	Hu.C1q.SIV _{mac239} gp140	Protection
4	Hu.C1q.SIV _{smE543} gp140	Risk

Reviewer #3

Major points:

The title of the paper is "Antibody Fab-Fc properties outperform titer in predictive models...", but the "titer" used in the analysis is not clearly defined. The manuscript states -- "Antibody quantity was also assessed at two different serum concentrations by an anti-IgG detection reagent". How do these two measured "titer" features relate to the titer from standard assays like ELISA or neutralization? What do the "high" and "low" features mean? Usually, the titer is defined by the assay and has just one measurement. So, here without explanation it is very confusing.

Authors named the titer feature as "quantitative", but with binary output of "high" and "low" it seems to be still rather qualitative or semi-quantitative.

Authors' response: We apologize for our lack of clarity regarding the "titer" measurement, and in particular the fact that we have two different features, from different serum concentrations, that are related to titer. A previous study (Brown et al, 2012) showed that anti-IgG detection reagent measurements are correlated with traditional ELISA-based measurements of titer. Hence we refer to Fc array features that use anti-IgG detection reagents as "titer" features. The two features with "high" and "low" sub-labels correspond to these measurements at two different *serum* concentrations. We have clarified our usage of this term and the measurements it describes in both the main text, when it is first introduced ("Antibody profiles" section, lines 142-161), and the Methods & Protocols ("Multiplexed IgG titering and Fc Array", lines 509-510).

The overall separation of the Detection Reagents into "qualitative" and "quantitative" is very confusing, because all Fc Array data were obtained, analyzed, and presented in the same manner, and all columns in Figures S1C,D have quantitative content.

Authors' response: The reviewer is entirely correct that all measurements are quantitative, and we apologize for the confusion regarding our characterization of some as "qualitative". By "qualitative", we mean that a feature has additional properties, or "qualities", beyond just magnitude. So whereas titer is only about magnitude, other Fc Array measurements capture additional characteristics of the antibodies (subclass, Fc receptor binding ability, etc.) that may be indicative of the degree to which a response is beneficial or detrimental. Since a high titer of a "bad" antibody could in fact lead to poor protection, our whole premise is that it is necessary to evaluate these qualities and not just assess titer, and we develop a framework that we show is able to leverage this to make better predictions. We have further edited the text throughout to try to eliminate the source of confusion and clarify the overarching philosophy regarding antibody qualities/properties. (The paragraph referenced above on "Antibody profiles", lines 142-161, has the most concentrated changes.)

The authors based their analysis on Fc Array data collected for a single timepoint. The manuscript says it is "post-vaccination, pre-challenge", but it would be useful if they can specify when after the boost the samples were collected. With 2 weeks between the boost and

challenge we may expect better identification of correlates of protection if samples were acquired just before the challenge and not right after the boost. If samples from different animals were taken at different time points within the two week-window, will it be taken into account by the models?

Authors' response: We thank the reviewer for pointing this out. The serum samples were all collected on the day of the challenge. We have now specified this timepoint in the initial description of the study (line 119). Since all samples were collected at the same time, we did not have to account for any timepoint differences.

Models based on humoral response alone fail to predict the protection outcome without addition of T cell response in some animals (Figure S5A). It is curious to see a discussion on which of the 4 humoral predictive features and to what extent were involved in these cases.

Authors' response: While we sympathize with the reviewer's interest in gaining insights into why this particular model worked better on some particular subjects than on others, we also don't want to read too much into these particulars based on small numbers. The three subjects were not strikingly different in one or more of these 4 features from the well-protected animals, hence the model's prediction of low risk. And of course it is likely that factors other than the humoral response relate to challenge outcomes (else we would expect all control animals to be infected uniformly). Unfortunately, though we consider the role of T cell responses to "explain" the poorer predictions among some animals, this was done based on the prior observation, as we consider the study to not be powered sufficiently well to support further sub-setting or sub-analysis, though it would be interesting to follow up on this suggestion if it were.

Very intriguing result of the manuscript is a high role of antibodies able to bind C1q in protection. Could it be indirectly linked to amount of generated T cell response? Did authors try to see the correlations between the three C1q features and IFN γ level? C1q has been reported to limit dendritic cell differentiation and activation by engaging LAIR-1 [Son et al. 2012], so we might expect some link.

Authors' response: The reviewer raises an interesting point. The simplest interpretation of potential mechanistic relevance of the C1q-associated correlate(s) is that antibody-dependent induction of the complement cascade may be involved in protection, for example, via direct viral lysis or the lysis of infected cells with envelope on their surfaces. However, there are many possible alternatives. To evaluate the possibility raised here, we looked for a relationship between these features and IFN γ , and observed no or weak correlation as shown in the table below, suggesting no such link to T-cell response.

	Fc Array Feature Name	Pearson's correlation coeff.
1	C1q.SIVmac1A11.gp140	-0.15
2	C1q.SIVsmH4.Gag	0.03
3	C1q.SIVmac239.Pol	0.31

Fouts et al. study have shown that addition of second adjuvant LTA1 to IL-12 reduced the high protection observed with IL-12 alone. It would be interesting to see a discussion if any of the specific analyzed Fc features or their combination may account for this effect.

Authors' response: We agree with the reviewer (and comment to that effect in the discussion) that it would be interesting to see what differences in response were caused due to the addition of LTA1 to IL-12. A group-specific survival analysis on IL-12 and LTA1+IL12 groups would provide a way to compare the humoral response between the two adjuvant groups. Unfortunately this study's sample size of 8 subjects per group is not sufficient to support this modeling approach. However, the four-way group classification result (Appendix Figure S7D) does show that none of the antibody Fc features specific to the LTA1+IL12 group appeared in the correlates of protection (Figure 2E&F).

Minor points:

The current description of experimental study [Ref.43] in Introduction is confusing due to numbering 1)-3). I suggest to remove 3) and talk about adjuvant in a separate sentence. Also, the Reader will benefit from overall more accurate description of the experiments here. For example, mentioning the route of prime vaccination and that the boost was with rhFLSC variant different from included in the prime vaccination.

Authors' response: We have changed the text (lines 99-109) to clarify these sources of confusion and generally make the protocol clearer.

suggestion for Table in Figure 1A to have just one unified description and not four same repeats in columns 3 and 4

Authors' response: We have made this clarifying change.

Fig.S5 description in the text "three of the six poorly protected animals that were predicted to have low risk of infection". suggestion to highlight those 6 animal points in different color to make it easy for Reader to follow.

Authors' response: We had mistakenly typed six instead of sixteen. We apologize for this error and thank the reviewer for catching it. We have now corrected the text (lines 263-266) and highlighted in the figure the three poorly predicted subjects with red circles. The sixteen subjects are shown in filled diamond shapes.

Fig.S5 A,C have "!" instead of "gamma"

Authors' response: On our machines the figure looks correct. We think this could be due to formatting incompatibility when importing .pdf figures into .doc files for the initial submission. Since the final figures will be in their original format, we hope the font problem will not persist.

2nd Editorial Decision

4th April 2019

Thank you again for sending us your revised manuscript. We are now satisfied with the modifications made and I am pleased to inform you that your paper has been accepted for publication.

Corresponding Author Name: Chris Bailey-Kellogg

Manuscript Number: MSB-18-8747